# Differential Expression of tRNA-Derived Small RNA Markers of Antidepressant Response and Functional Forecast of Duloxetine in MDD Patients

**DOI:** 10.3390/genes16020162

**Published:** 2025-01-27

**Authors:** Xiaoyan Wang, Ming Gao, Jing Song, Miaolong Li, Yu Chen, Yingfang Lv, Wei Jia, Bingbing Wan

**Affiliations:** 1Key Laboratory of Systems Biomedicine (Ministry of Education), Shanghai Center for Systems Biomedicine, Shanghai Jiao Tong University, Shanghai 200240, China; cathywxy@sjtu.edu.cn (X.W.);; 2School of Life Sciences and Biotechnology, Shanghai Jiao Tong University, Shanghai 200240, China; 3School of Medicine, Renji Hospital, Shanghai Jiao Tong University, Shanghai 200240, China; 4School of Chemical Science and Engineering, Tongji University, Shanghai 200070, China; 5Department of Clinical Medicine, He University, Shenyang 110163, China; 6Department of Pharmacology and Pharmacy, University of Hong Kong, Hong Kong, China

**Keywords:** duloxetine, functional prediction, small non-coding RNAs, tRNA-derived small RNAs, biomarkers, ECM1, BAFF

## Abstract

Background/Objectives: Duloxetine, despite being a leading treatment option for major depressive disorder (MDD), exhibits a relatively low adequate response rate when used as a monotherapy, and the fundamental molecular mechanisms remain largely elusive. tRNA-derived small RNA (tsRNA) is a particularly interesting and new class of molecules that is becoming increasingly noticeable for investigation. Methods: We integrated small RNA sequencing with bioinformatics approaches to dissect the expression profiles of tsRNAs and decipher their functional roles post-duloxetine treatment. Subsequently, molecular docking experiments were carried out to validate the potential functions. Results: Ten tsRNAs significantly changed in the duloxetine response group after an 8-week therapy. Correlation analyses revealed that these tsRNAs predominantly interacted with miRNAs across multiple biological pathways and processes, such as the ECM-receptor interaction and B cell activation. Molecular docking analysis corroborated the binding capabilities of duloxetine with key proteins associated with ECM1 and BAFF, respectively. Conclusions: The identified changes in tsRNAs can precisely mirror the response of duloxetine in MDD treatment, offering novel insights into the underlying mechanisms of duloxetine action.

## 1. Introduction

Major depression represents a heterogeneous ailment, characterized by a multiplicity of symptoms, including cognitive impairments and diverse forms of physical disabilities. This condition imposes significant health and social burdens on a global scale, as attested by numerous studies [1]. Among the array of treatment modalities available, such as psychological behavior intervention and diet and nutrition intervention, drug therapy remains the cornerstone of treatment. Antidepressants have demonstrated pronounced efficacy in treating moderate to severe depressive episodes; nevertheless, the individual responses to these pharmacological interventions exhibit substantial variability. Despite the wide range of antidepressants in clinical use today, a staggering 30–40% of patients fail to achieve a full therapeutic response [2,3]. The lack of response not only prolongs patients’ suffering but also places an onerous burden on both patients and their families. Thus, delving deeper into the factors that influence antidepressant treatment outcomes is of paramount importance and urgency.

Duloxetine, being one of the most prevalently prescribed antidepressants, has drawn considerable clinical attention regarding its therapeutic efficacy and has been the subject of in-depth investigations [4,5]. Gene sequencing studies have revealed genetic variants upstream of STAC1 that are correlated with the treatment response in patients with depression, whether they were administered duloxetine or a placebo [6]. Non-coding RNAs (ncRNAs), a category of RNA molecules that do not participate in protein translation, function as regulatory elements, akin to molecular spinners and switches, modulating the transcriptional activity of gene expression. Dysregulation of ncRNAs frequently disrupts the biochemical pathways implicated in major depressive disorder (MDD) [7,8]. The identification of changes in ncRNAs in response to drugs used for treating MDD holds the promise of uncovering biomarkers capable of predicting drug responses, thereby facilitating the efficient screening of patients who exhibit minimal or no response to traditional medications [9].

Transfer RNAs (tRNAs), a class of ncRNAs that rank second in abundance within cells, can be enzymatically cleaved into a diverse assortment of ncRNA fragments, typically ranging from 18 to 40 nucleotides in length. These fragments, generated from either precursor tRNAs or mature tRNAs through the action of specific endonucleases, have recently emerged as functionally significant small non-coding RNAs, termed tsRNAs. Mounting evidence implicates multiple tsRNA dysregulations in various human diseases [10,11,12]. Notably, tsRNAs have been reported to modulate pathophysiological alterations in neurological disorders, such as neurodegeneration and nerve injury, suggesting their potential role in regulating mental disorders [13]. Current research indicates that specific tRNA-modifying enzymes and tsRNAs could serve as promising diagnostic biomarkers and therapeutic targets [14]. However, to date, no studies have explored the role of tsRNAs in human depression.

In the present study, we sought to investigate tsRNAs as prospective biomarkers for antidepressant response by employing small-RNA sequencing on matched specimens from patients with MDD. These patients were participants in a placebo-controlled, randomized trial evaluating duloxetine treatment, with samples collected both prior to treatment initiation and eight weeks after the commencement of therapy. Our findings demonstrate that the levels of ten tsRNAs are differentially regulated in relation to antidepressant response and are involved in modulating genes associated with crucial biological processes, including extracellular matrix (ECM)-receptor interaction, the transforming growth factor-β (TGF-β) signaling pathway, fatty acid biosynthesis, thyroid hormone synthesis, the Hippo signaling pathway, the plasma membrane signaling receptor complex, and humoral immune response. Further molecular docking analysis confirmed the binding potential of duloxetine and key proteins, verifying the pathways involved in the drug response of duloxetine involved in the above tsRNA.

## 2. Materials and Methods

Dataset: The miRNA-seq sequencing and miRNA expression files were collected from GSE97154 [15]. This study is a double-blind clinical trial that is registered at www.ClinicalTrials.gov (11984A NCT00635219). A total of 258 patients (males *n* = 80; females *n* = 178) were enrolled and diagnosed with MDD. Participants were randomly designated to obtain either a placebo or 60 mg of duloxetine. Peripheral blood samples were collected at the beginning of the study and following the treatment period. For inclusion criteria, patients were aged from 19 to 74 and were diagnosed with MDD and a major depressive episode (MDE) lasting more than three months, having a severity score on the Montgomery–Asberg Depression Rating Scale at baseline of no less than 22. For exclusion criteria, patients had undergone at least two prior antidepressant (AD) treatments, experienced electroconvulsive therapy within the six weeks preceding the study or had a major depressive episode (MDE) along with bipolar disorder, psychotic features, or a recent substance use disorder. The percentage change of Montgomery–Asberg Depression Rating Scale (MADRS) scores was calculated (from week 0 to week 8 therapy) to quantify the therapy response. The responder/non-responder were categorized according to a great decrease in Montgomery–Asberg Depression Rating Scale scores from week 0.

Sequencing data analysis: All sequencing data were sequenced on the HiSeq2500 Illumina sequencer (Illumina, San Diego, CA, USA). The Cutadpt 2.1 was utilized, and low-quality reads were filtered [16]. The expression data of tsRNAs were obtained after clean reads were aligned to the mature-tRNA genome using MINTmap(v2.0) [17]. Missing values were imputed by MetImp 1.2 [18].

Statistical analysis: We conducted a comparison of treated patients (week 8) and baseline (week 0) using the student *t*-test. Correlation analysis between the single vectors was performed using the Spearman correlation. For two matrix correlation analysis, the Spearman correlation first analyzed the correlation analysis between the single tsRNA with single miRNA, and then miRNAs with *p* < 0.001 were selected for the matrix Mantel test. DIANA tools were used for miRNA pathway analysis [19]. R software (version 4.0.2) was employed for all analyses.

Target Prediction: The RNAhybrid algorithm was employed to forecast the potential binding mRNAs’ targets, using a screening criteria of energy <−25 kcal/mol) (https://bibiserv.cebitec.uni-bielefeld.de/, accessed on 28 January 2024). Shingo was applied to analyze cellular components and biological processes, as well as identify potential functions (http://bioinformatics.sdstate.edu/go/, accessed on 28 January 2024).

Molecular docking: The crystal structure of the key protein (ECM1 and BAFF) was obtained in the Protein Data Bank (PDB, https://www.rcsb.org/), respectively. The 3D structures of duloxetine were downloaded from PubChem (https://pubchem.ncbi.nlm.nih.gov/). The Autodock 4.0 was applied to perform molecular docking and calculate binding affinity. Each calculation generated 50 structures, and the molecular docking output was prioritized according to the frequency of possible ligand-binding sites and free-energy score. The docking results of ECM1 proteins and duloxetine were visualized by PyMOL 2.2.0 software.

## 3. Results

### 3.1. Differential tsRNA Expression After Duloxetine Therapy

#### 3.1.1. The Workflow of the Study

The study began by obtaining sequencing data in fastq format from the GSE97154 dataset. The sequencing reads were then processed to remove adapters, ensuring high-quality data for subsequent analysis. The workflow of the study is shown in Figure 1.

#### 3.1.2. Differential tsRNA Expression Identification

From the processed data, transfer RNA-derived small RNAs (tsRNAs) were extracted for further investigation. After we extracted the tsRNA expression data from small-RNA-sequencing blood samples, we compared the expression of tsRNAs after and before therapy (week 8 to week 0). A 5% false discovery rate (FDR) using the Benjamini–Hochberg correction for multiple testing was applied in differential analysis. The findings indicate a differential expression of ten tsRNAs in the duloxetine response group after an 8-week treatment period (Table 1) and two tsRNAs in the placebo non-responsive group (Table 2). We analyzed each group’s overlap through the Venn diagram and found that tRF-36-D4ZWRNU3KQ9MV1B overlapped in the duloxetine response and placebo non-responsive groups (Figure 2). We further analyzed the expression of ten tsRNAs significantly changed in the duloxetine response group before and after treatment and found that ten tsRNAs cluster into two principal classes in the heatmap (Figure 2). These tsRNAs with *p* value < 0.05 were calculated in a paired student *t*-test and listed (Appendix A).

### 3.2. Correlation Analysis and Functional Enrichment Analysis

#### 3.2.1. Correlation Analysis of tsRNAs and tsRNA Expression

To analyze the relationship between these tsRNAs, we applied Spearman correlation analysis. The heatmap in Figure 3 shows that these tsRNAs are divided into two categories, each of which is positively correlated internally. Following an 8-week treatment, we observed that among the two types of tsRNA, four were upregulated, and six were downregulated (box plots in Figure 3). Unlike miRNAs that are widely studied, only a few tsRNA functions are known. Here, we use the correlation analysis of tsRNAs and miRNAs, generated by previous research, and enhance the functions of these miRNAs, which are strongly related to tsRNAs, to predict the functions attributed to them.

#### 3.2.2. Correlation Analysis Between Significant tsRNAs and miRNAs

Figure 4A illustrates the correlation analysis between tsRNAs and miRNAs, highlighting specific interactions that could play crucial roles in regulating the molecular pathways implicated in MDD. This figure reveals patterns of connectivity that suggest both direct and indirect regulatory relationships, affecting the transcriptional landscape in response to duloxetine treatment. Particularly, we can observe that that tRF.50.PNR8YPLON4VN1EH6KK8 and tRF.33.86V8WPMN1E8Y0E have more interaction pathways with miRNAs, indicating their potentially influential roles in regulating or being regulated by the expression of various miRNAs.

#### 3.2.3. Correlation Analysis miRNA Function Study

The results of the correlated miRNA function study are shown in Figure 4B. miR-146a and miR-146b were correlated with thyroid hormone synthesis. The ECM-receptor interaction was correlated with miR-425, while fatty acid biosynthesis was correlated with miR-16, respectively.

We selected miRNAs with Mantel’s *p* value less than 0.01 for enrichment analysis. Enrichment analysis revealed that these miRNAs mainly interact with ECM-receptor interaction together with fatty acid biosynthesis, thyroid hormone synthesis, TGF-β, and the Hippo signaling pathway (Appendix A).

### 3.3. Bioinformatic Prediction of the Ten Significantly Expressed tsRNAs

The function of ten significantly expressed tsRNAs in the duloxetine response group was studied using bioinformatic techniques. Figure 5A depicts the enrichment analysis of the biological process. Among them, the notable enrichment and the significant terms discovered were, respectively, the plasma membrane signaling receptor complex and the humoral immune response in molecular function. The cellular component of tsRNA target genes is shown in Figure 5B with the identical findings.

### 3.4. Duloxetine with Its Corresponding Proteins

According to the result of functional enrichment analysis, ECM1, an important protein in ECM-receptor interaction, was proposed to find a potential relationship with duloxetine. Molecular docking analysis displayed the possibility that duloxetine binds to GLU199 of ECM1. The predicted binding energy is −4.84 kcal/mol (Figure 6A), which indicates that there is a strong affinity between protein and ligand. By observing the protein surface model, one can also see that the ligand is only attached to the protein surface, and a hydrogen bond is formed between the ligand and the protein residue, which fully demonstrates the interaction between duloxetine and the ECM1 protein.

According to biological process enrichment analysis and cellular component enrichment of tsRNA target genes (Figure 5), B cell activation is the most frequent biological process. Hence, BAFF (B cell activating factor) is selected. Our docking results show that the interaction free energy between BAFF and duloxetine is −4.44 kcal/mol, which indicates that there is a strong affinity between protein and ligand. By observing the protein surface model, we found that small molecules stick to the protein surface and zoom in on the area. We found that the structure formed on the protein surface is similar to that of small molecules, which is conducive to inducing small molecules to bind to it. At the same time, we also showed the unit structure between protein and ligand and found that hydrogen bonds are formed between leucine at position 272 and the ligand, further stabilizing the complex.

## 4. Discussion

Duloxetine, a prominent member of the class of selective serotonin and norepinephrine reuptake inhibitors (SNRIs), has carved out a significant niche in clinical practice for the management of major depressive disorder (MDD) [20,21], whereas SSRIs, like Fluoxetine, Paroxetine, and Sertraline, primarily act on the reuptake of 5-HT. Our choice was based on duloxetine’s dual serotonin-norepinephrine reuptake inhibition mechanism and its established clinical efficacy in treating major depressive disorder (MDD) [21]. It was also demonstrated to possess neuroprotective effects, likely via these pathways, in addition to its capacity to modify neurotransmitter signaling [5]. At cellular level, emerging evidence reports that alterations in autophagy, inflammation, coke death, and apoptosis pathways can potentially influence its development and progression [22]. Given this intricate web of cellular events and their far-reaching implications, it becomes eminently pertinent to explore the possibility of identifying reliable biomarkers.

The employment of miRNA biomarkers to monitor patient responses in mood disorders marks a transformative leap in treatment paradigms. Multiple studies have indicated that interventions like SNRIs, SSRIs, serotonin modulation, and electroconvulsive therapy (ECT) achieve their therapeutic impacts, in part, by targeting miRNAs [23], clearly suggesting miRNAs’ potential as powerful indicators for gauging treatment efficacy.

In mice with CUMS-induced depression, miR-134 and miR-124a were markedly elevated in the frontal lobe and hippocampus, but these levels declined post-duloxetine treatment [24,25]. Clinically, there are discernible differences in miRNA expression between patients who respond to duloxetine and those who do not. Here, miR-16, miR-146a, and miR-21p have been identified as promising markers linked to remission under duloxetine treatment [15]. On the other hand, multiple miRNAs have been linked to either a response to treatment or a heightened risk of major depression. These significant findings were validated that downregulated miR-146a, miR-24, miR-425, and miR-3074 after treatment were strongly correlated, indicating a common mode of action [15].

By delving into miRNA profiles, we meticulously examined the alterations in miRNA expression triggered by duloxetine treatment. Our findings strongly suggest that duloxetine’s therapeutic function may well be intertwined with its influence on miRNA expression patterns, thereby affirming its role as a modulator of miRNA levels, which aligns with existing literature [26].

Enrichment analysis unveiled that tsRNA-related miRNAs partake in critical biological processes such as ECM-receptor interaction, thyroid hormone synthesis, fatty acid biosynthesis, the TGF-β signaling pathway, and the Hippo signaling pathway. Meanwhile, extensive research has explored the connections between fatty acid biosynthesis, the TGF-β signaling pathway [27], thyroid hormone synthesis, and depression [28].

Investigations have spotlighted the fact that the extracellular matrix (ECM) serves as a vital conduit for communication, potentially influencing behavior stress regulation and depression [29,30], with the intertwined connection between the ECM and immune processes [31,32]. The ECM has also been definitively shown to hold a crucial role in orchestrating inflammatory and neuropathic pain, as corroborated by multiple studies [33,34,35]. Widely acknowledged in the medical field, duloxetine exhibits remarkable efficacy not only in alleviating the depressive symptoms of patients with major depressive disorder (MDD) but also in substantially ameliorating diverse forms of pain [36,37]. However, the precise mechanism underlying its pain-relieving capabilities remains elusive. Our molecular docking experiments yielded a clear and potent indication of duloxetine’s interaction with ECM1, a key protein within the ECM framework (Figure 6A). This discovery not only provides robust support for earlier enrichment analysis outcomes but also posits that the therapeutic impact of duloxetine on MDD symptoms could be intertwined with its binding to the ECM. This binding, potentially implicating immune-related pathways, offers tantalizing clues about the drug’s latent mechanisms for pain regulation.

Simultaneously, we carried out an in-depth analysis of the tsRNA-miRNA function concerning duloxetine’s effects. Pathway enrichment analysis highlighted the plasma membrane signaling receptor complex and the humoral immune response as the most prominent terms in molecular function and biological processes. Among them, the B cell activation pathway emerges as a focal point of interest. In light of this, BAFF (B cell activating factor), a cytokine that plays a pivotal role in activating B cells, belonging to the tumor necrosis factor (TNF) ligand family, was chosen for molecular docking experiments with duloxetine. The outcomes were quite revealing, demonstrating a notable binding affinity between the two entities (Figure 6B). BAFF is closely related to autoimmunity and immune regulation [38,39], thereby making its interaction with duloxetine a potentially crucial aspect in understanding the drug’s immunomodulatory potential and therapeutic implications.

Given the findings regarding interactions with BAFF, ECM1, and TGF-β, it would be relevant to discuss the role of neuroinflammation in depression. Duloxetine was reported to possess anti-inflammatory (decreasing TGF-β proteins) and antioxidant properties to regulate the expression of angiogenesis and neurotrophic factors [40], which may relate to neuroinflammation and the broader immunological mechanisms underlying depression. This will tie in with emerging evidence of SSRIs exhibiting properties beyond serotonin reuptake inhibition, which could provide a more holistic understanding of their therapeutic effects.

tRF-36D4ZWRNU3KQ9MV1B was retained in the analysis, and its significance was interpreted within the broader context of depression and antidepressant mechanisms. This overlap does not diminish its importance but rather underscores the complexity of tsRNA-mediated regulation in MDD. Further experimental validation is suggested to unravel its precise functional role in these overlapping conditions.

Actually, one tsRNA has been proven to serve as a key target in depression, and the silencing of it diminishes the occurrence of ferroptosis and safeguards neurons from injury [41]. Significant downregulation of tsRNA was evident after an 8-week treatment course and functioned as a promising baseline predictor of a patient’s response to antidepressant therapy [42]. Consistently, our results also suggest that tsRNA serves as a predictive biomarker for the drug treatment effect of major depressive disorder, indicating that specific tsRNAs present in peripheral blood show a significant response to depressive disorders and their symptoms, and the underlying mechanism is worthy of in-depth exploration.

The limitation of this study lies in the fact that it only explored data from public databases without performing multi-center validation. The sequencing results need to be further verified in other samples from multiple clinical centers in the future. Moreover, the results of the molecular docking also demand further molecular biological experimental validation to confirm that the drug duloxetine has the actual ability to bind to and even regulate the receptor target. Although this study has completed the expression profile and functional prediction, there remain questions that require further exploration. Additionally, experimental verification both in vitro and in vivo is needed to identify the functions of the candidate tsRNAs and also the related signaling pathways mentioned above. This will provide valuable guidance in elucidating the antidepressant mechanism and even in developing new indications of duloxetine.

## 5. Conclusions

Our study has successfully demonstrated that the alterations in tsRNA expression patterns can accurately reflect the response of duloxetine in MDD treatment. This not only provides novel insights into the long—elusive molecular mechanisms of duloxetine action but also paves the way for developing more accurate diagnostic and prognostic tools.Moreover, the molecular docking analysis validating duloxetine’s binding with key proteins related to ECM1 and BAFF enriches our understanding of its therapeutic mechanism. These findings are expected to stimulate further research on the complex interactions between tsRNAs and other cellular components. Such research may lead to innovative therapeutic strategies, enhancing the effectiveness of duloxetine or other antidepressants, thus potentially alleviating the burden of MDD.

## Figures and Tables

**Figure 1 genes-16-00162-f001:**
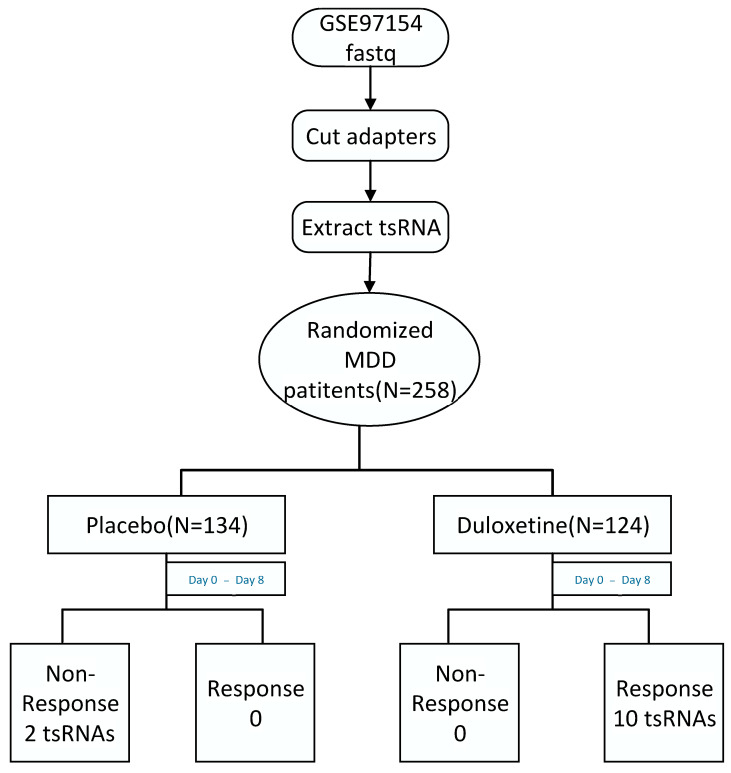
Workflow of data process.

**Figure 2 genes-16-00162-f002:**
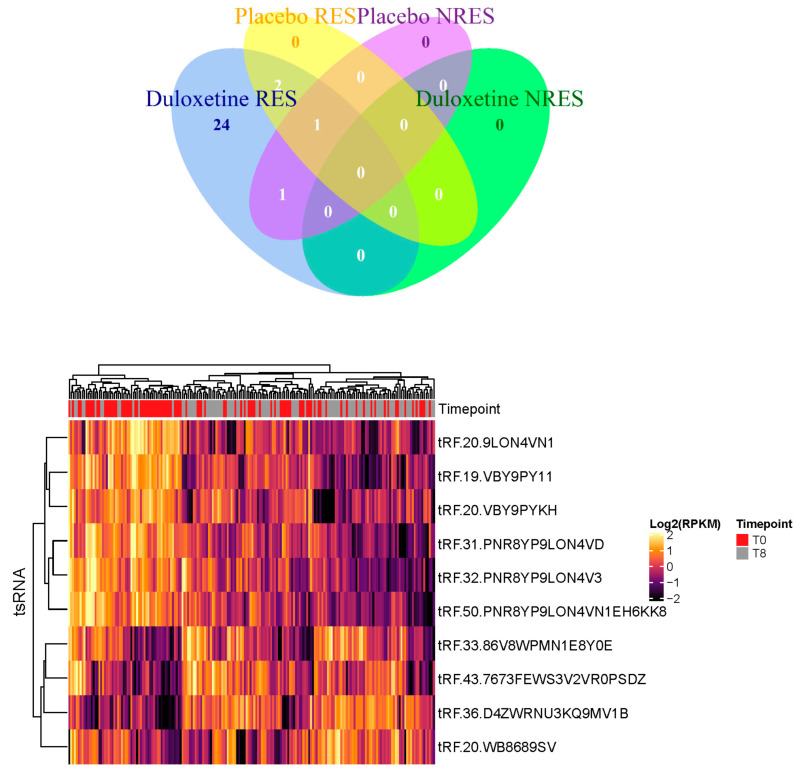
Venn diagram of each group and duloxetine treatment response tsRNA expression analysis. Unsupervised hierarchical clustering of all significant tsRNA markers for duloxetine treatment response. Each row is tsRNA, and the column is the patient sample.

**Figure 3 genes-16-00162-f003:**
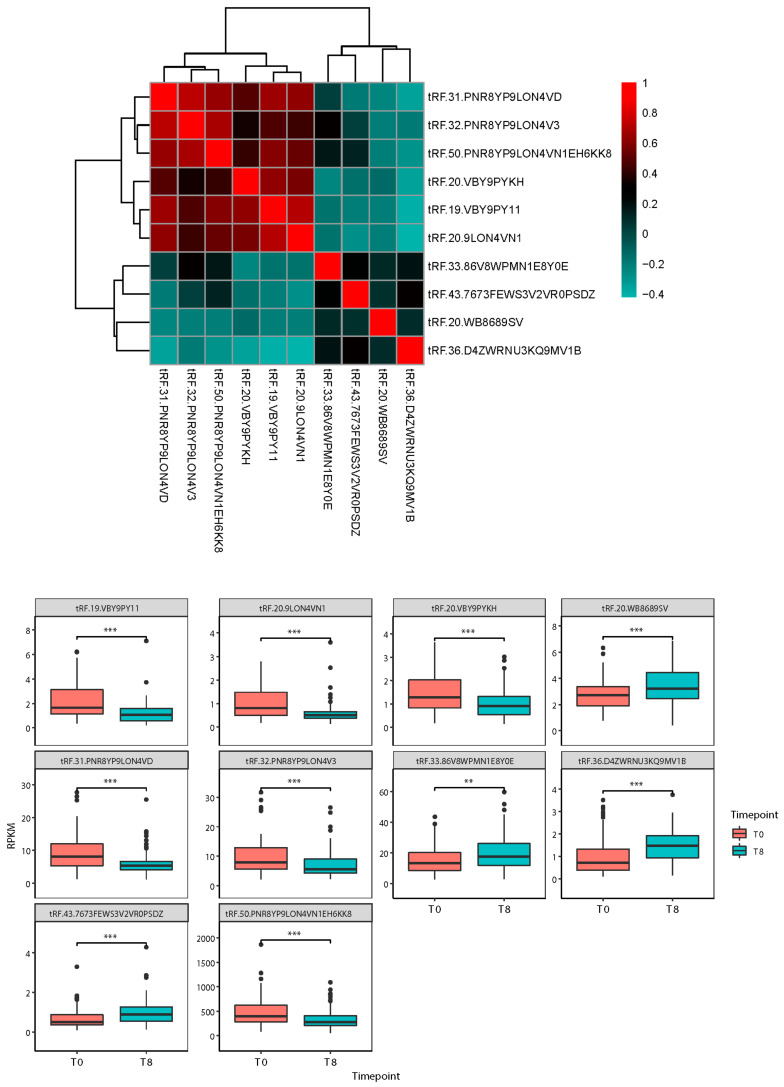
Correlation analysis of tsRNAs and tsRNA expression data. (** *p* value less than 0.01, *** *p* value less than 0.001).

**Figure 4 genes-16-00162-f004:**
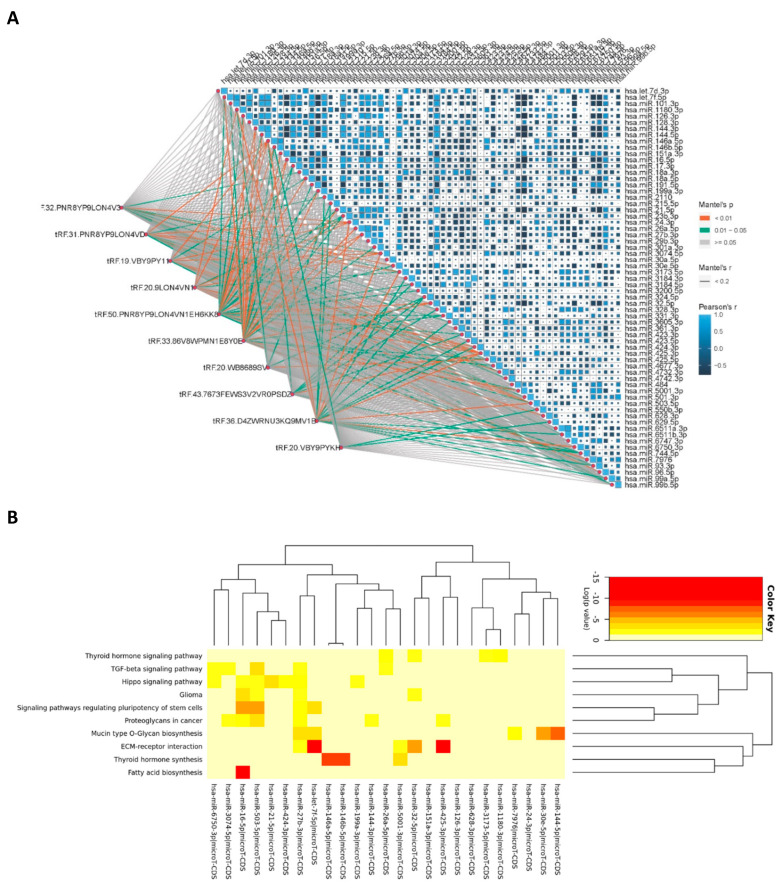
Correlation analysis between significant tsRNAs and miRNAs (**A**) and correlation analysis miRNA function study (**B**).

**Figure 5 genes-16-00162-f005:**
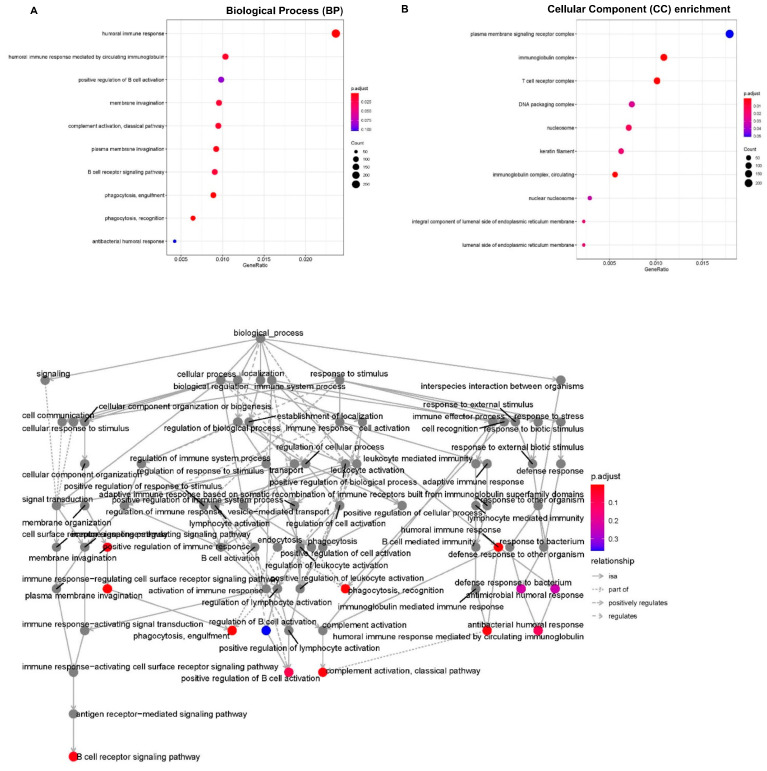
Biological process enrichment analysis and cellular component enrichment of tsRNA target genes. (**A**) The top 10 biological processes are presented in the bubble chart and in the whole network. (**B**) The top 10 cellular component enrichments are presented in the bubble chart (The area of the circle indicates the tsRNA target gene number).

**Figure 6 genes-16-00162-f006:**
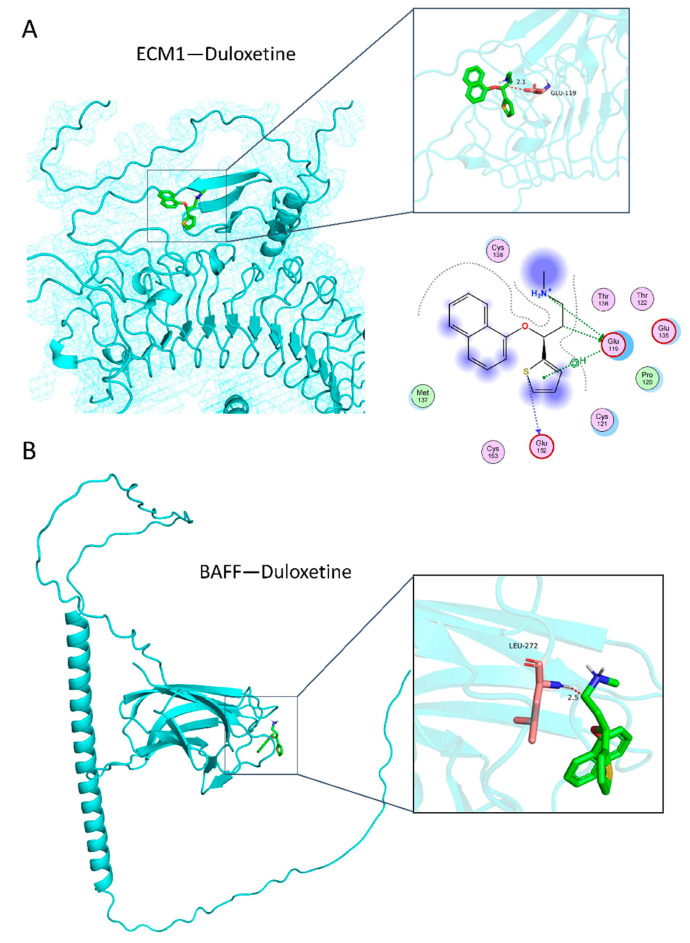
The 3D docking conformations of ECM1 and duloxetine (**A**); BAFF and duloxetine (**B**).

**Table 1 genes-16-00162-t001:** Duloxetine response Signiant tsRNAs (FDR < 0.001).

Name	Sequence	*p* Values	Fold Change	FDR
tRF-20-9LON4VN1	TGGTAGAATTCTCGCCTGCC	2.65 × 10^−9^	0.567917175	1.08 × 10^−6^
tRF-31-PNR8YP9LON4VD	GCATTGGTGGTTCAGTGGTAGAATTCTCGCC	2.02 × 10^−8^	0.655811181	2.73 × 10^−6^
tRF-19-VBY9PY11	TAGAATTCTCGCCTGCCAC	1.75 × 10^−8^	0.542665077	2.73 × 10^−6^
tRF-50-PNR8YP9LON4VN1EH6KK8	GCATTGGTGGTTCAGTGGTAGAATTCTCGCCTGCCACGCGGGAGGCCCGG	4.47 × 10^−8^	0.684414445	4.53 × 10^−6^
tRF-32-PNR8YP9LON4V3	GCATTGGTGGTTCAGTGGTAGAATTCTCGCCT	2.62 × 10^−6^	0.744936648	0.000178
tRF-20-WB8689SV	TCGAATCCCATCCTCGTCGC	2.98 × 10^−6^	1.259833712	0.000178
tRF-36-D4ZWRNU3KQ9MV1B	AAGTGTTTGTGGGTTTAAGTCCCATTGGTCTAGCCA	3.08 × 10^−6^	1.489983712	0.000178
tRF-20-VBY9PYKH	TAGAATTCTCGCCTGCCACG	4.19 × 10^−6^	0.705437268	0.000213
tRF-33-86V8WPMN1E8Y0E	TCCCATATGGTCTAGCGGTTAGGATTCCTGGTT	2.02 × 10^−5^	1.319922659	0.00082
tRF-43-7673FEWS3V2VR0PSDZ	GTTCAGTGGTAGAATTCTCGCCTGCCACGCGGGAGGCCCGGGT	1.85 × 10^−5^	1.493597202	0.00082

**Table 2 genes-16-00162-t002:** Placebo non-response Signiant tsRNAs (FDR < 0.001).

Name	Sequence	*p* Values	Fold Change	FDR
tRF-34-10I9BZBZOS4YE2	AGGAGATTTCAACTTAACTTGACCGCTCTGACCA	1.82 × 10^−7^	1.617784657	7.38 × 10^−5^
tRF-36-D4ZWRNU3KQ9MV1B	AAGTGTTTGTGGGTTTAAGTCCCATTGGTCTAGCCA	5.62 × 10^−7^	1.697292931	0.000114

## Data Availability

The original contributions presented in this study are included in the article/Appendix A. Further inquiries can be directed to the corresponding authors.

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
