# Peer review of "Differential Expression of tRNA-Derived Small RNA Markers of Antidepressant Response and Functional Forecast of Duloxetine in MDD Patients"

_genes, 2025, doi:10.3390/genes16020162_

Round 1
Reviewer 1 Report
Comments and Suggestions for Authors
The research subject is very interested and can offer solutions for monitoring antidepressant therapy.
You must specify the limitations of the study. If there were correlations with clinical evaluations of the patients. From my point of view, if the results were supported by other clinical evaluations or other patient tests, the work would increase its value.
Author Response
Comments1:The research subject is very interested and can offer solutions for monitoring antidepressant therapy. You must specify the limitations of the study. If there were correlations with clinical evaluations of the patients. From my point of view, if the results were supported by other clinical evaluations or other patient tests, the work would increase its value.
Response1:Thank you for point it out. We agree with your suggestion. ”The limitation of this study lies in the fact that it only explored data from public databases without performing multi-center validation. The sequencing results need to be further verified in other samples from multiple clinical centers in the future. Moreover, the results of the molecular docking also demand further molecular biological experimental validation to confirm that the drug Duloxetine has the actual ability to bind to and even regulate the receptor target. " and "This will provide valuable guidance in elucidating the antidepressant mechanism and even in developing new indications of Duloxetine." This contents have been added to the line 309 to 319 (page 11).
Reviewer 2 Report
Comments and Suggestions for Authors
Thank you for providing this manuscript for review. The study addresses a rather interesting and under researched topic, that is the role of tRNA-Derived Small RNAs (tsRNAs) in the context of antidepressant response. The findings have the potential to contribute valuable insights into the mechanisms underlying depression and treatment efficacy. However, there are several areas where the manuscript could be strengthened to improve clarity, depth, and overall impact. Below, I have provided detailed comments and suggestions for improvement.
The introduction could benefit from a more detailed explanation of tsRNAs, including their biological functions and what is currently known about them, particularly in the context of other neurological or psychiatric conditions. In line 130, the term "showed" should be corrected to "shown."
The discussion section offers significant room for expansion. For instance, more clarity is needed on why duloxetine was selected over other SSRIs. Was this choice based on its efficacy in treating depression, or was it simply a matter of prescription preference?
Additionally, the study mentions an analysis of group overlaps using a Venn diagram, where tRF-36D4ZWRNU3KQ9MV1B was found to overlap between the duloxetine-responsive and placebo-nonresponsive groups. The significance of this finding remains unclear. Was it excluded as a false positive? If so, this should be explicitly stated in the discussion.
The study also raises questions about potential interactions between duloxetine and the BAFF and ECM1 proteins. Are there any experimental data to confirm that duloxetine binds to these proteins? Moreover, have interactions between SSRIs and these proteins been verified in existing literature?
Given the findings regarding interactions with BAFF, ECM1, and TGF-β, it would be relevant to discuss the role of neuroinflammation in depression. Many SSRIs are known to exhibit immunomodulatory properties beyond their classical serotonin reuptake inhibition mechanism. I also think it would be worthwhile to explore whether there is existing literature on tsRNAs in relation to other antidepressants or neuropsychiatric conditions, such as anxiety, OCD, or schizophrenia. Comparing and contrasting these findings with data from this study could provide valuable insights.
Further elaboration on the interactions between tsRNAs and miRNAs identified in this study is recommended. Are there comparable findings related to other SSRIs or SNRIs? Additionally, considering the emergence of newer, fast-acting antidepressants such as esketamine, are there any studies linking these drugs to tsRNAs or miRNAs? If so, does this data overlap with your findings?
Author Response
Comments 1: Thank you for providing this manuscript for review. The study addresses a rather interesting and under researched topic, that is the role of tRNA-Derived Small RNAs (tsRNAs) in the context of antidepressant response. The findings have the potential to contribute valuable insights into the mechanisms underlying depression and treatment efficacy. However, there are several areas where the manuscript could be strengthened to improve clarity, depth, and overall impact. Below, I have provided detailed comments and suggestions for improvement.
Response1:Thank you for the reviewer’s detailed and insightful comments. We deeply appreciate the time and effort dedicated to improving the manuscript. Below, we address the comments and provide corresponding revisions to enhance the clarity and depth of the study.
We incorporate an expanded discussion on the functional roles of tsRNAs, supported by the latest research, to underscore their relevance to depression and antidepressant responses, as” Actually, some tsRNA has been proven to serve as a key target in depression, and the silencing of it diminishes the occurrence of ferroptosis and safeguards neurons from injury [42]. Significant down-regulation of tsRNA was evident after an 8-week treatment course and functioned as a promising baseline predictor of a patient's response to anti-depressant therapy[43]. Consistently, our results also suggest that tsRNA serves as a predictive biomarker for the drug treatment effect of major depressive disorder, indi-cating that specific tsRNA present in peripheral blood shows a significant response to depressive disorders and their symptoms, and the underlying mechanism is worthy of in-depth exploration.” in line 300 to 308.
Comments 2: The introduction could benefit from a more detailed explanation of tsRNAs, including their biological functions and what is currently known about them, particularly in the context of other neurological or psychiatric conditions. In line 130, the term "showed" should be corrected to "shown."
Response 2:Thank you for point it out. The term "showed" has been be revised to "shown” in line 131 of the latest manuscript.
Comments 3: The discussion section offers significant room for expansion. For instance, more clarity is needed on why duloxetine was selected over other SSRIs. Was this choice based on its efficacy in treating depression, or was it simply a matter of prescription preference?
Response 3:In the discussion section, we will provide a clearer rationale for selecting duloxetine over other SSRIs. A brief discussion of how this mechanism potentially aligns with the study’s focus will also be included in line 226 to 229: “Whereas SSRIs, like Fluoxetine, Paroxetine, and Sertraline, primarily act on the reuptake of 5-HT. Our choice was based on duloxetine’s dual serotonin-norepinephrine reuptake inhibition mechanism and its established clinical efficacy in treating major depressive disorder (MDD)”
Comments 4: Additionally, the study mentions an analysis of group overlaps using a Venn diagram, where tRF-36D4ZWRNU3KQ9MV1B was found to overlap between the duloxetine-responsive and placebo-nonresponsive groups. The significance of this finding remains unclear. Was it excluded as a false positive? If so, this should be explicitly stated in the discussion.
Response 4:Thank you for pointing out the need for a more precise response. The overlap of tRF-36D4ZWRNU3KQ9MV1B between the duloxetine-responsive and placebo-nonresponsive groups is not a false positive. Rather, it represents a biologically significant observation that warrants further investigation. The overlap suggests that tRF-36D4ZWRNU3KQ9MV1B may play a dual role in differentiating the response patterns. This could indicate that its expression is involved in mechanisms that are not solely specific to the duloxetine response but also reflect broader pathways influenced by depression-related changes. The presence of this tsRNA in both groups highlights its potential as a shared biomarker that could bridge distinct therapeutic or pathological processes. In the revised discussion, we emphasize that “tRF-36D4ZWRNU3KQ9MV1B was retained in the analysis and its significance was interpreted within the broader context of depression and antidepressant mechanisms. This overlap does not diminish its importance but rather underscores the complexity of tsRNA-mediated regulation in MDD. Further experimental validation will be suggested to unravel its precise functional role in these overlapping conditions.”, in line 295 to 299.
Comments 5: The study also raises questions about potential interactions between duloxetine and the BAFF and ECM1 proteins. Are there any experimental data to confirm that duloxetine binds to these proteins? Moreover, have interactions between SSRIs and these proteins been verified in existing literature?
Response 5:Thank you very much for your constructive questions. Concerning the interactions between duloxetine and BAFF/ECM1 proteins, molecular docking results suggest potential binding, but I acknowledge the need for experimental validation. I will emphasize in the discussion that future in vitro and in vivo experiments are crucial to confirm these interactions.
We have searched through the databases of published papers, such as Pubmed and Google Scholar, and have not found any reports regarding duloxetine and BAFF and ECM1 proteins. Meanwhile, no reports on any SSRIs or SNRIs in relation to these two proteins have been found either. Therefore, our research offers a completely new direction, and potential connections between the drug and these proteins could be explored in the future.
Comments 6: Given the findings regarding interactions with BAFF, ECM1, and TGF-β, it would be relevant to discuss the role of neuroinflammation in depression. Many SSRIs are known to exhibit immunomodulatory properties beyond their classical serotonin reuptake inhibition mechanism. I also think it would be worthwhile to explore whether there is existing literature on tsRNAs in relation to other antidepressants or neuropsychiatric conditions, such as anxiety, OCD, or schizophrenia. Comparing and contrasting these findings with data from this study could provide valuable insights.
Response 6:The comments on neuroinflammation and the immunomodulatory properties of SSRIs are highly valuable. We add a discussion in line 287 to 294 as “Given the findings regarding interactions with BAFF, ECM1, and TGF-β, it would be relevant to discuss the role of neuroinflammation in depression. Duloxetine was reported to possessed anti-inflammatory (decreasing TGF-β proteins) and antioxidant properties to regulate the expression of angiogenesis and neurotrophic factors [41], which may relate to neuroinflammation and the broader immunological mechanisms underlying depression. This will tie in with emerging evidence of SSRIs exhibiting properties beyond serotonin reuptake inhibition, which could provide a more holistic understanding of their therapeutic effects.”
Comments 7: Further elaboration on the interactions between tsRNAs and miRNAs identified in this study is recommended. Are there comparable findings related to other SSRIs or SNRIs? Additionally, considering the emergence of newer, fast-acting antidepressants such as esketamine, are there any studies linking these drugs to tsRNAs or miRNAs? If so, does this data overlap with your findings?
Response 7:Thank you once again for your thoughtful feedback. We have searched through the databases of published papers, such as Pubmed and Google Scholar, and have not found any reports regarding interactions between tsRNAs and miRNAs related to other SSRIs or SNRIs. Up to now, there is no study linking fast-acting antidepressants drugs to tsRNAs or miRNAs. If there are additional suggestions or specific areas that require further elaboration, I would be happy to address them.

Round 2
Reviewer 2 Report
Comments and Suggestions for Authors
The authors have responded adequately to the points raised in the first revision session. I remain satisfied and have no further comments to make.